# Co-exposure risks of pesticides residues and bacterial contamination in fresh fruits and vegetables under smallholder horticultural production systems in Tanzania

Jones A. Kapeleka[1,2]*, Elingarami Sauli[1], Omowunmi Sadik[3], Patrick A. Ndakidemi[1]

**1** The Nelson Mandela African Institution of Science and Technology (NM-AIST), Arusha, Tanzania,
**2** Tropical Pesticides Research Institute (TPRI), Arusha, Tanzania, **3** Department of Chemistry and Environmental Science, New Jersey Institute of Technology, Tiernan Hall Newark, Newark, New Jersey, United States of America

* jak78tz@gmail.com

**Data Availability Statement:** All relevant data are within the paper and its Supporting Information files.

## Abstract

This study was carried out to investigate the risks of simultaneous exposure to pesticide residues and bacteria contaminants in locally produced fresh vegetables and vegetables in Tanzania. A total of 613 samples were analyzed for pesticide residues, out of which 250 were also analyzed for bacterial contamination. Overall, 47.5% had pesticide residues, 74.2% exceeded Maximum Residue Levels (MRLs). Organophosphorus (95.2%), organochlorines (24.0%), pyrethroids (17.3%), and carbamates (9.2%) residues dominated. MRL values were mostly exceeded in tomatoes, onions, watermelons, cucumbers, Chinese cabbage, and sweet paper. Tetramethrin (0.0329–1.3733 mg/kg), pirimiphos-methyl (0.0003–1.4093 mg/kg), permethrin (0.0009–2.4537 mg/kg), endosulfan (beta) (0.0008–2.3416 mg/kg), carbaryl (0.0215–1.5068 mg/kg), profenofos (0.0176–2.1377 mg/kg), chlorpyrifos (0.0004–1.2549 mg/kg) and dieldrin (0.0011–0.5271 mg/kg) exceeded MRLs. The prevalence of bacteria contamination was high (63.2%). *Enterobacter* (55.6%) *Pseudomonas aeruginosa* (32.4%), *E. coli* (28.2%), *Citrobacter* (26.8%), *Klebsiella oxytoca* (14.8%), and *Salmonella* (7.7%) were isolated. Furthermore, 46.4% tested positive for both pesticide residues and bacterial contaminants. Vegetables from farms (60.7%) contained more dual contaminants than market-based vegetables (41.8%). This may have resulted from excessive pesticide use and unhygienic handling of fresh fruits and vegetables at production level. Binary logistic regression showed that fresh fruits and vegetables with pesticide residues were 2.231 times more likely to have bacteria contaminants (OR: 2.231; 95% CI: 0.501, 8.802). The contamination levels of pesticide residues and bacterial contaminants could be perceived as a serious problem as most fresh fruits and vegetables recorded values of pesticide residues far above the MRLs with pathogenic bacteria isolated in higher proportions. MRLs was higher in most vegetables consumed raw or semi-cooked such as watermelons, carrots, cucumber, tomatoes, onion and sweet paper. There is an urgent need to develop pesticide monitoring and surveillance systems at farmer level, educating farmers and

**Funding:** The grant was received by JAK. The study was funded by Tanzania Horticultural Association (TAHA) in collaboration with Rikolto in Tanzania. The funders had no role in study design, data collection and analysis, decision to publish, or preparation of the manuscript.

**Competing interests:** The authors have declared that no competing interests exist.

promoting the use of greener pesticides to mitigate the health effects of pesticides and bacterial contaminants.

## Introduction

The safety of locally produced and consumed horticultural crops raises serious public health concerns due to dietary exposure effects and the occurrence of foodborne diseases [1]. Fresh fruits and vegetables constitute the main food and cash crop among many smallholder farmers in Tanzania [2]. Unacceptable pesticide residue levels and bacterial contamination of these vegetables resulting from poor agricultural practices, food handling, and unhygienic factor increases dietary exposure risks [3,4].

Consumption of fresh vegetables is encouraged by governmental health agency campaigns as an important part of a healthy diet due to their protective health effects against non-communicable diseases and as a source of important micronutrients [4,5]. However, pesticide residues and bacterial pathogens are considered to be the most important food safety issue for fresh produce [5]. Vegetables that are mostly eaten raw or semi-processed are at higher risk for contamination by pesticide residues and pathogenic foodborne microbes [6]. These vegetables can be contaminated with both bacteria and pesticide residues during cultivation, harvesting, and transportation [7].

In order to ensure food safety and improve quality values for horticultural crops, Maximum Residue Levels (MRLs) limits must be adhered to [8], but meeting these international food safety requirements had been a significant challenge for the fresh produce in many African countries [5]. Exposure to pesticides is, therefore, more likely to affect the general population through the consumption of contaminated food with pesticide residues [5].

Pesticides residues have been found in food materials from both developed and developing countries [4,9–11], but a higher prevalence of residues exceeding the MRLs are observed in food products from developing countries. These foods exceeding the tolerance of pesticide residues in foods are consumed by the general public, pausing public health concern due to increased risk of detrimental effects of pesticide exposure. [12,13].

The adverse effect of pesticides used may also affect soil fertility because of the adverse effects of pesticidal chemicals on soil microorganisms [14]. Pesticide sorption can either enhance or decrease microbial degradation rates in soil [15]. Their catabolic gene and respective enzymes are capable of degrading pesticides in highly contaminated areas [16,17], indicating a symbiotic association and risks of mutual occurrence of pesticides and microbial contaminants in fresh fruits and vegetables.

Microbiological and chemical analyses on fresh produce had been, therefore, reported to be an important step in the control of fresh vegetable contamination [5]. This is because pesticide use had been linked to support and increase microbial growth in vegetable produce [18]. Despite this fact, few studies on the effects of co-contamination of fresh vegetables and the resultant health risks had been reported [1,6]. Paradoxically, no such studies had been reported in Tanzania, and the available data elsewhere did not provide comparable levels of simultaneous occurrence and proportions of pesticides and bacterial contaminants in vegetable samples. The rationale for assessing the levels of pesticide residues and bacterial contamination was to establish the risks of simultaneous exposure to harmful pesticide and bacterial contaminants through the consumption of locally produced fresh vegetables and to derive factors influencing the co-occurrence of pesticides and bacterial contaminants. This study, therefore,

assessed co-exposure risks and levels of pesticide residues and bacterial contaminants in locally produced and consumed vegetables in Tanzania.

## Materials and methods

### Study area

The assessment was undertaken in the highest smallholder vegetable production and consumption zones in Tanzania. These included southern highlands (Morogoro and Iringa), northern corridor (Arusha, Kilimanjaro, and Manyara) and coastal zone (Dar es Salaam).

### Samples

A total of 613 samples from 17 horticultural crops produced by smallholder farmers were analyzed for pesticide residues, out of which 250 were also tested for microbial contamination. Samples were randomly purchased from market places, farmer fields during harvesting period and along with highway selling points. A subsample of 250 was analyzed for bacterial contamination due to the homogeneity of samples. Likewise, only samples collected from areas where samples could reach the lab within 4 hours of sample collection were considered for the assessment of bacterial contamination. Inclusion criteria were based on the most consumed vegetables. Samples were taken in sterile polyethylene bags which were placed in iced packed cool boxes and transported to the laboratory. Samples were selected according to FAO recommended guidelines on sampling for pesticide residue analysis, method of sampling for the determination of pesticide residues where each whole fruit, vegetable, or a bunch of vegetables were taken to form a unit, except where these were very small. The minimum weight for the small and medium-size sample was 1kg and that for large size product was 2 kg.

### Chemicals and reagent

Fifty-two (52) certified pesticide standards with purities between 99.0–99.9% were obtained the official registrant companies of specific pesticides in Tanzania. Tetramethrin standard was obtained from Star Import & Export, Tanzania, Pirimiphos–methyl, Permethrin, Profenofos, and Lambda Cyhalothrine from Syngenta Crop Protection Ag, Switzerland; Cabaryl from Bayer Environmental Science, Triadimefon from Meru Agro Tours & Consultant Tanzania, Dimethoate from Sapa Chemical Industrial Ltd, Chlorpyrifos and Bifenthrin from Balton Tanzania Ltd-Tanzania, Oxyfluorfen from Bayer East Africa Ltd-Kenya, Malathion from Tanzania Crop Care Limited, Tanzania while Propanil was obtained Suba Agro-Trading and Engineering Co.Ltd–Tanzania. Solvents at HPLC grade, including acetonitrile, acetic acid, and acetone, salts of analytical grade such as anhydrous magnesium sulphate, sodium sulphate, and sodium acetate; Primary Secondary Amine (PSA), were locally sourced from Immuno Lab Supplies, Dar es Salaam Tanzania.

### Standard preparation for pesticides residues

Individual pesticide standard stock solutions were prepared in acetone to a final concentration of 20.0 mg/L and stored at -20 ˚C. The standard mixed component solution was then prepared by diluting each primary standard solution with acetonitrile with 1% acetic acid (1:10 v/v ml). This was then used for spiking extracted fruit and vegetable samples. A 100μl of 1 mg/ml heptachlor was used as an internal standard to ensure the accuracy of the GC-MS response.

## Sample preparation and extraction for pesticides residues analysis

Sample preparation and extraction were done following the Association of Official Analytical Chemists (AOAC) Official Method (QuEChERS Protocol), an extraction method for pesticide residues in foods by acetonitrile extraction and partitioning with magnesium sulfate, applicable for pesticides in fruits and vegetables [19,20]. Fresh fruits and vegetables analyzed for pesticide residues included onions, watermelons, tomatoes, sweet paper, Chinese cabbage, African nightshade, carrots, amaranths, kale, Ethiopian mustered, African eggplants, eggplant, green beans, cabbage, and okra. Samples were processed soon after collection and transported to the laboratory at 4˚C until they were analyzed.

Briefly, a whole, unwashed 200g sample was homogenized through chopping into small pieces, grinding, and blending. This is because, according to the CODEX Guidelines on Good Practice in Pesticide Residue Analysis, pesticides may tend to collect in the stem area of fruits and on the top of vegetables. Therefore vertical sections must be cut through the stem and centre of fruits, and the top and center of vegetables should be chopped and homogenized for analysis [21]. Exactly 15g of the sample was transferred in a 50ml centrifuge tube, and 15ml of ethyl acetate was added. An internal standard, 100μl, was added and vortexed for 1min, then 5g $MgSO_4$ and 1.5g sodium acetate were added, vortexed to 1 min, and centrifuged at 4000 r.p.m for 10min. A supernatant layer (3ml) was transferred in a 15ml centrifuge tube, 300mg $MgSO_4$, and 150mg Primary and Secondary Amine (PSA) added, vortexed for 1min and centrifuged again at 4000 r.p.m for 10min. A final supernatant layer (1.5ml) was transferred into G.C. vials and injected in GC-MS for the detection of pesticide residues.

## Gas chromatography-mass spectrometry system and operating conditions

The Agilent 7890A Gas Chromatography-Mass Spectrometer (GC-MS), which is equipped with 7693 auto-sampler coupled with a 7000B triple quadrupole M.S. system was used in the detection and in quality assurance of pesticides residues. A fused silica DB35 capillary column, 30mm long with 0.25 mm internal diameter and 0.25 μm film operating at a range of 50 ˚C to 360 ˚C was used with the internal temperature set at 50˚C for 1 minute, constantly raised to 150 ˚C at a rate of 50˚C per minute, followed by 280 ˚C at a heating rate of 5 ˚C per minute and held for four minutes. The injector temperature was 250˚C, and a carrier gas was helium (99.9%) at a flow rate of 1.2 ml $min^{-1}$ splitless injection. The injection volume was 1μl at a pressure of 43.193 Psi. The MS ion source temperature was 250˚C operated in full scan mode at a scan range of 50–550˚C atomic mass unit.

## Assessment of bacterial contamination

From the 613 fresh fruits and vegetable samples collected, 250 were portioned aseptically in sterilized sampling bags and transported to the lab in sterilized cool boxes within 4 hours and processed soon after arrival. Samples were homogenized using a pre-sterilized blender, and the sample mixture was filtered through a filter paper to get a clear filtrate. About 5mls of the filtrate was inoculated in the tryptic soy broth (TSB) enrichment broth and incubated for 24hrs. After growth, each sample was streaked onto selective and differential agar plates [MacConkey and Xylose Lactose Deoxycholate (XLD)] agar and incubated for 24hrs [22]. Pure bacteria colonies were isolated and sub-cultured in nutrient agar (N.A.), tryptic soy agar (TSA) and incubated for the other 24hrs at a temperature of 35˚C as explained by Ruangpan and Tendencia [23]. Identification of the bacterial strains was made using biochemical identification tests for common gram-negative bacteria isolates including Simmons Citrate Agar, Lysine Iron Agar, Urea Agar Base, Triple Iron Agar, and Sulphur Indoor Motility Agar [22,24].

## Statistical and data analysis

Statistical analysis of data was done using SPSS 22.0 computer software. Descriptive statistics such as frequencies, percentages, mean, and standard deviations were performed. Chi-square ($\chi$2) test was used to determine statistically significant differences in pesticide residues and bacterial contamination rates across different variables. Pearson correlation test was used to establish the correlation between the levels of pesticide and bacterial contamination. Binary logistic regression analysis was done to determine factors influencing the likelihood of simultaneous pesticide and bacterial contamination of vegetables under smallholder vegetable production systems in Tanzania. A significant level for the results was accepted at $p < 0.05$.

# Results

## Recovery, quantitative evaluation and detection limits

The method performance for the quantification of the concentration of pesticide residues in fresh fruits and vegetables widely produced and locally consumes was validated according to the European Commission guidelines for pesticide analysis [21]. This was done by determining recoveries, Limit of Detection (LOD), Limit of Quantification (L.O. Q), precision, and linearity. Recovery was performed by analyzing a mixture of standard pesticides in blank vegetable samples at different known concentrations of 0.1 mg/kg, 0.5mg/kg, 1mg/kg, and 1.5mg/kg in triplicates. A 15 g homogenized sample was spiked with pesticide mixture standard solution and allowed to equilibrate for 3 hours prior to extraction. Extraction and analysis were done according to the procedures described previously. Calibration curves constructed from the concentration and peak areas of the chromatograms obtained with standards were used to calculate recovery values. The mean recoveries ranged between 75% and 115%, with an average of 94%. Precision was determined by calculating the relative standard deviation (RSD) of the lowest concentration that could show linearity in blank vegetable samples. The relative standard deviations (RSD) obtained was below 10% with an average of 7.7%.

Linearity was determined by analyzing a mixture of pesticide standards at different concentrations ranging from 0.005–0.02mg/kg. The area of the corresponding peak in the sample was then compared with that of the standard. Specificity and validity of the method was monitored by running control blank vegetable samples simultaneously, in which no chromatographic peak was observed at the same retention times of target pesticides which indicated non occurrence of interferences. Analyses were carried out in triplicates and the mean concentrations based on the number of samples that tested positive for each sample calculated. Limits of quantification for the method were calculated by considering a value 10 times that of background noise while detection limits were found by determining the lowest concentrations of the residues in each of the matrices that could be reproducibly measured at the operating conditions of the G.C. using a signal-to-noise (S/N) ratio of 3. The calculated limit of detection limits (LOD) ranged from 0.002–0.006 mg/kg, while the Limits of Quantification (L.O. Q) ranged from 0.002 to 0.016.

## Pesticides residues in fresh fruits and vegetable samples

A total of 613 samples were collected and analyzed for pesticide residues. These vegetable samples represent the most common vegetables produced by smallholder farmers and mainly consumed locally within the production areas and transported to nearby urban and peri-urban markets. A considerable proportion (47.5%) of all samples tested for pesticide residues had at least one detectable pesticide residue. Samples from farms (46.3%) and markets (46.8%) had comparable proportions of detectable levels of pesticides while a large proportion of vegetable

**Table 1. Pesticides residues tests by sampling places.**

| Variable | | Sampling place | | | | | | Total | |
|---|---|---|---|---|---|---|---|---|---|
| | | Farm | | Market | | Highway | | n | % |
| | | n | % | n | % | n | % | | |
| Pesticides residues test | No pesticides residues detected | 132 | 53.7 | 177 | 53.2 | 13 | 38.2 | 322 | 52.5 |
| | Pesticides residues detected | 114 | 46.3 | 156 | 46.8 | 21 | 61.8 | 291 | 47.5 |
| Total | | 246 | 100.0 | 333 | 100.0 | 34 | 100.0 | 613 | 100.0 |

samples sold along highways (61.8%) had detectable pesticide residues (Table 1). Chi-square test showed no statistically differences (p = 0.22) in the levels of contamination. Samples from Dar es Salaam (73.7%), Morogoro (70.3%), and Iringa (70.1%) recorded the highest proportions of pesticide residues compared with samples from Manyara (17.9%), Kilimanjaro (33.6%) and Arusha (38.5%).

## Classes of pesticides used and residues detected from fresh vegetable

Fourteen (14) pesticide classes were used in smallholder vegetable production systems. These included organophosphates (96.6%), carbamates (56.1%), and substituted benzene (35.6%). Mixed formulations were also used, including a combination of pyrethroid and organophosphates (28.8%), carbamate, and acylalanine (22.9%). Others include avermectin (28.1%), and dithiocarbamate (19.5%), inorganic fungicides, pyrethroids, and organochlorines (Table 2). Furthermore, 13 different pesticide classes were also detected from fresh vegetable samples as pesticides residues. Organophosphorus pesticide residues were detected in 95.2% of all vegetable samples with detectable levels of pesticide residues while organochlorine was detected in 24%, pyrethroids in 17.3%, and carbamates in 9.2%. Pesticides uses in smallholder vegetable production, therefore, accounts for the accumulation of pesticides residues in fresh fruits and vegetables.

## Occurrence of multiple pesticides residues

Multiple pesticide residues (up to seven in a single sample) were detected from analyzed samples. Table 3 shows that 49.1% of all vegetable samples had one pesticide residue, 31.1% had

**Table 2. Classes of pesticides used and residues detected from fresh vegetable.**

| Chemical families of pesticides used | n | % | Chemical families of pesticides residues detected | n | % |
|---|---|---|---|---|---|
| Organophosphorus | 285 | 96.6 | Organophosphorus | 258 | 95.2 |
| Carbamate | 158 | 56.1 | Organochlorine | 65 | 24 |
| Substituted benzen | 101 | 35.6 | Pyrethroid | 47 | 17.3 |
| Pyrethroid+Organophosphorus | 84 | 28.8 | Carbamates | 25 | 9.2 |
| Avermectin | 82 | 28.1 | Diphenyl Ether | 23 | 8.5 |
| Carbamate+Acylalanine | 67 | 22.9 | Organic fungicides | 16 | 5.9 |
| Dithiocarbamate | 57 | 19.5 | Sulfur | 15 | 5.5 |
| Inorganic fungicide | 39 | 13.4 | Thiophthalimides | 7 | 2.6 |
| Pyrethroid | 26 | 8.9 | Nitrophenolic and Nitrocresolic herbicides | 3 | 1.1 |
| Organochlorine | 26 | 8.9 | Substituted benzene | 3 | 1.1 |
| Pyrethroid+Nitroimidazolidylideneamine | 25 | 8.6 | Conazole | 2 | 0.7 |
| Oxadiazines | 12 | 4.1 | Anilides | 1 | 0.4 |
| Conazole | 6 | 2.1 | Chlorophenoxy compounds | 1 | 0.4 |
| Propionic acid | 5 | 1.7 | | | |

**Table 3. Number of pesticides residues detected by sampling places.**

| Variable | | Sampling place | | | | | | Total | |
|---|---|---|---|---|---|---|---|---|---|
| | | Farm | | Market | | Highway | | n | % |
| | | n | % | n | % | n | % | | |
| Number of residues | One pesticide residue detected | 48 | 44.9 | 79 | 51.0 | 12 | 57.1 | 139 | 49.1 |
| | Two pesticides residues detected | 33 | 30.8 | 47 | 30.3 | 8 | 38.1 | 88 | 31.1 |
| | Three pesticides residues detected | 16 | 15.0 | 16 | 10.3 | 1 | 4.8 | 33 | 11.7 |
| | Five pesticides residues detected | 6 | 5.6 | 5 | 3.2 | | | 11 | 3.9 |
| | Four pesticides residues detected | 4 | 3.7 | 5 | 3.2 | | | 9 | 3.2 |
| | Six pesticides residues detected | | | 2 | 1.3 | | | 2 | 0.7 |
| | Seven pesticides residues detected | | | 1 | 0.6 | | | 1 | 0.4 |
| Total | | 107 | 100.0 | 155 | 100.0 | 21 | 100.0 | 283 | 100.0 |

two detectable pesticide residues, and 11.7% had three different types of pesticides residues. Chi-square test (p = 0.762) showed that samples from all sampling places had comparable numbers of pesticides residues detected.

## Types of pesticides residues detected from vegetable samples

Fifty-two (52) different types of pesticide residues were detected from fresh fruits and vegetable samples. From all samples tested, oxyfluorfen was detected from 12.2% (55.6% of tomatoes, 27.6% onions, and 11.1% sweet paper), lambda cyhalothrine from 10.1% (60% of tomatoes, 20% onions, and 13.3% sweet paper) and profenofos from 9.5% (42.9% tomato, 35.7% onion, and 14.4% sweet paper). Triadimenol was detected from 8.8%, (46.2% tomato, 15.4% onion and 23.1% sweet paper), chlorpyrifos, from 8.1% (33.3% tomato and onion respectively and 16.7% sweet paper), cyhalothrin (Gamma) from 8.1% (33.5% tomato, 25% onion and 8.3% sweet paper) and triadimefon from 8.1% (16.7% tomato, 58.3% onion and 8.3% sweet paper). Pirimiphos–methyl was detected from 7.4% (45.5% tomato and 54.5% onion), Endosulfan (Beta) from 6.1% (44.4% tomato and onion respectively), and Carbofuran from 6.1% (44.4% tomato and onion respectively). Dieldrin was detected from Chinese cabbage and sweet paper (20% respectively) and 30% onion. Samples of representative chromatograms of the detected pesticides are presented in Figs 1, 2, 3, 4 and 5.

## Concentration of pesticides residues detected

Some fresh fruits and vegetables recorded average high mean concentration of pesticides residues. Tetramethrin (Mean; Range) (0.7031; 0.0329–1.3733 mg/kg), pirimiphos–methyl (0.6798; 0.0003–1.4093mg/kg), dinoseb acetate (0.3235; 0.2492–0.8961mg/kg), permethrin (0.8488; 0.0009–2.4537mg/kg), endosulfan (beta) (0.3457; 0.0008–2.3416mg/kg), cabaryl (0.3178; 0.0215–1.5068mg/kg), triadimefon (0.3122; 0.0011–1.6047mg/kg) and profenofos (0.2701; 0.0176–2.1377mg/kg) had high residue concentrations. Other pesticides with high mean residue concentration were acephate, lambda cyhalothrine, trichlorform, dimethoate, chlorpyrifos, triadimenol, oxyfluorfen and dieldrin (Table 4). In tomato samples, chlorpyrifos recorded an average of 0.2612 mg/kg, onion (0.1539mg/kg), and okra (0.1224mg/kg) while dieldrin recorded 0.1971mg/kg in tomatoes, 0.1887mg/kg in African eggplants, 0.1387mg/kg Chinese cabbage, and 0.1404mg/kg in watermelons.

## Pesticides residue excess over Codex MRL default limit

Onions (2.4537 mg/kg), watermelons (1.3733 mg/kg), tomatoes (1.2549 mg/kg) and sweet paper (1.5068 mg/kg) recorded the highest maxima pesticides residues among all fresh fruits

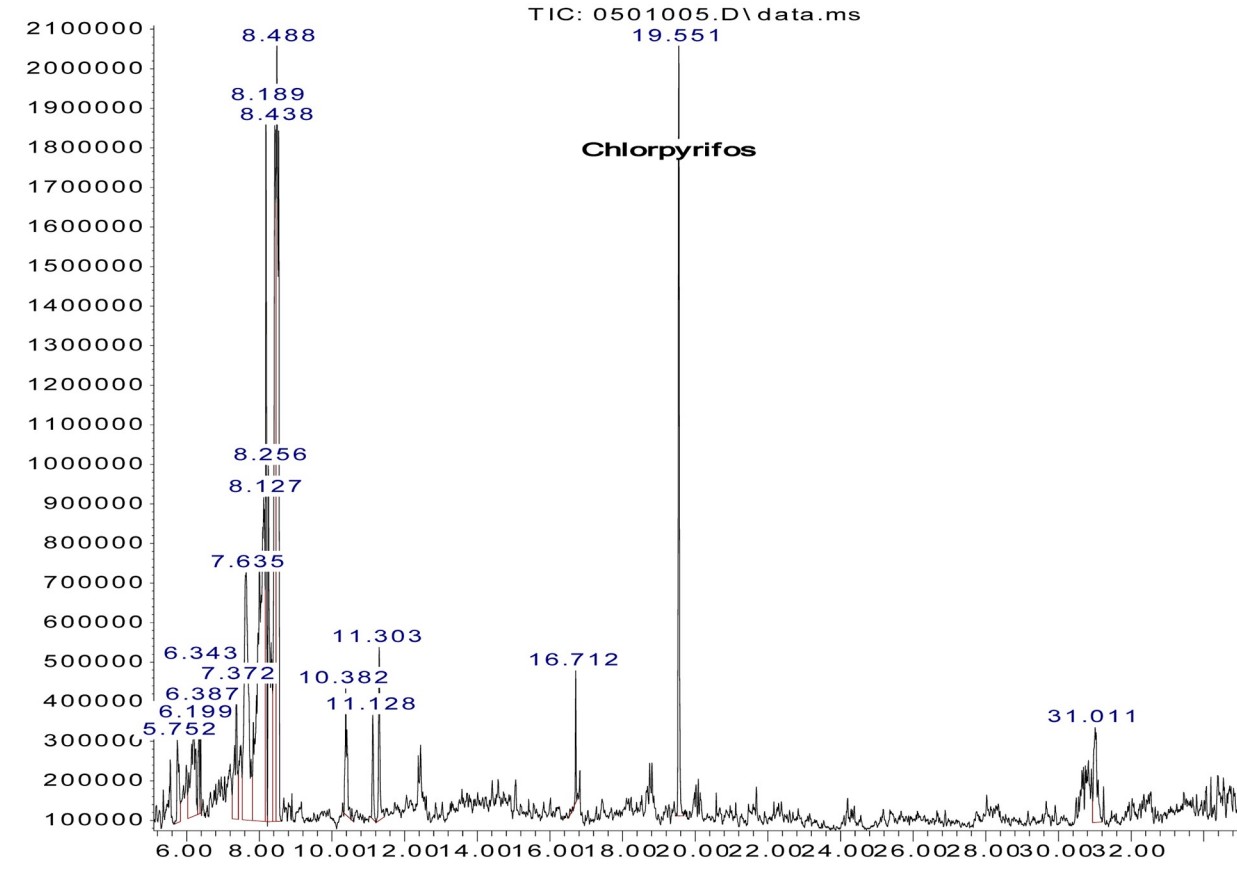

**Fig 1. Chromatogram of chloropyrifos in Chinese cabbage.**

and vegetable samples analyzed for pesticides residues. Consequently, the same recorded the highest excess levels of pesticide residues over a default codex maximum pesticide residue limit of 0.01mg/kg of food sample (Table 5).

## Levels of pesticides residues by sampling sites

About three quarters (74.2%) of vegetable samples with detectable pesticide residues had residue levels above Codex MRL standards. Vegetable samples from farms and market had comparable high levels of pesticide residues (71.0% and 74.7% respectively), while samples from highways recorded the highest (90.0%) proportion of pesticides above Codex MRL standards (Table 6).

## Bacterial contaminants of fresh fruits and vegetables

**Prevalence of bacterial contamination in fresh fruits and vegetable.** The bacterial contamination prevalence was relatively high, as 63.2% of all fresh fruits and vegetable samples tested positive for pathogenic bacterial contamination. The highest bacterial contamination was found in kale (88.2%), cabbage (81.8%), spinach (77.8%), Ethiopian mustard (73.7%), water splashed on vegetables, and nightshade (71.4%) respectively, and amaranths and

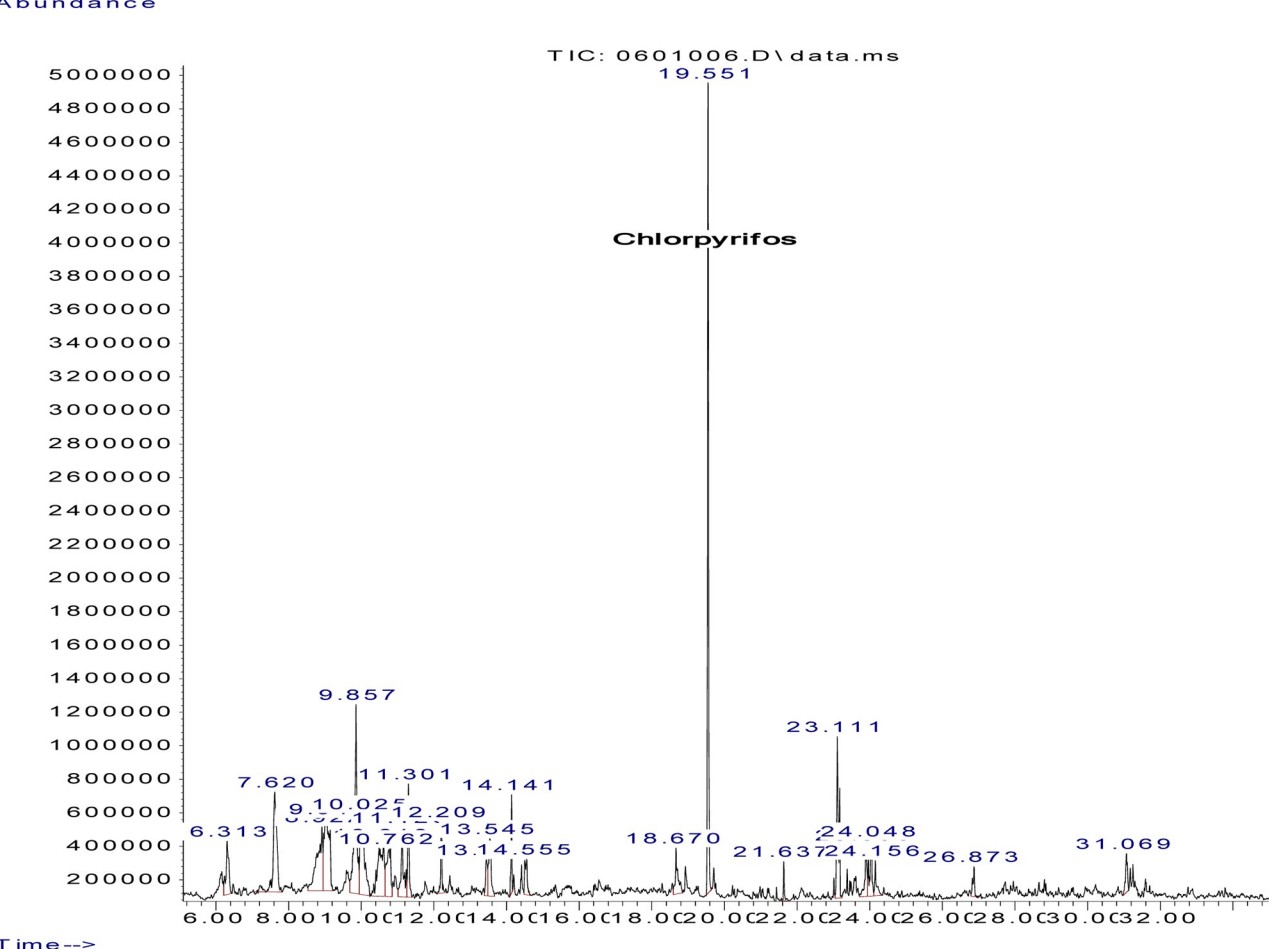

**Fig 2. Chromatogram of chloropyrifos in watermelon.**

watermelons (66.7%) respectively. Moreover, tomatoes (61.1%), onions (64.7%) carrots, ripe banana (57.1%) and Chinese cabbage (54.5%) had considerably high pathogenic contaminations. The least contaminated were okra and avocadoes (33.3%) respectively.

Six pathogenic bacteria were isolated from fresh fruits and vegetables. These included *Enterobacter* (55.6%) *Pseudomonas aeruginosa* (32.4%), *E. coli* (28.2%), *Citrobacter* (26.8%), *Klebsiella oxytoca* (14.8%), and *Salmonella* (7.7%). Samples from farms recorded higher prevalence of *E. coli* contamination (45.8%) compared to the markets samples (24.6%) while *Salmonella* was isolated from samples collected from the market places only (Table 7).

## Pathogenic bacterial strains isolated by vegetable types

Kale (73.3%), watermelons (58.8%) African eggplant (55.6%), and water used to freshen vegetables (44.0%) were highly contaminated with *Enterobacter*, while spinach (60%), cabbages (50%) and tomatoes (41.7%) were contaminated with *Pseudomonas aeruginosa*. Furthermore kale (26.7%), Chinese cabbage (23.5%) and tomatoes (16.7%) were contaminated with *E.coli*. *Salmonella* strains were isolated from onion (26.7%), amaranths (20%), and Chinese cabbage (14.7%), and water samples (4.0%). *Citrobacter* was isolated from cucumbers (33.3%), nightshade (29.6%), onion (26.7%), and sweet paper (20%). *Klebsiella oxytoca* was found in

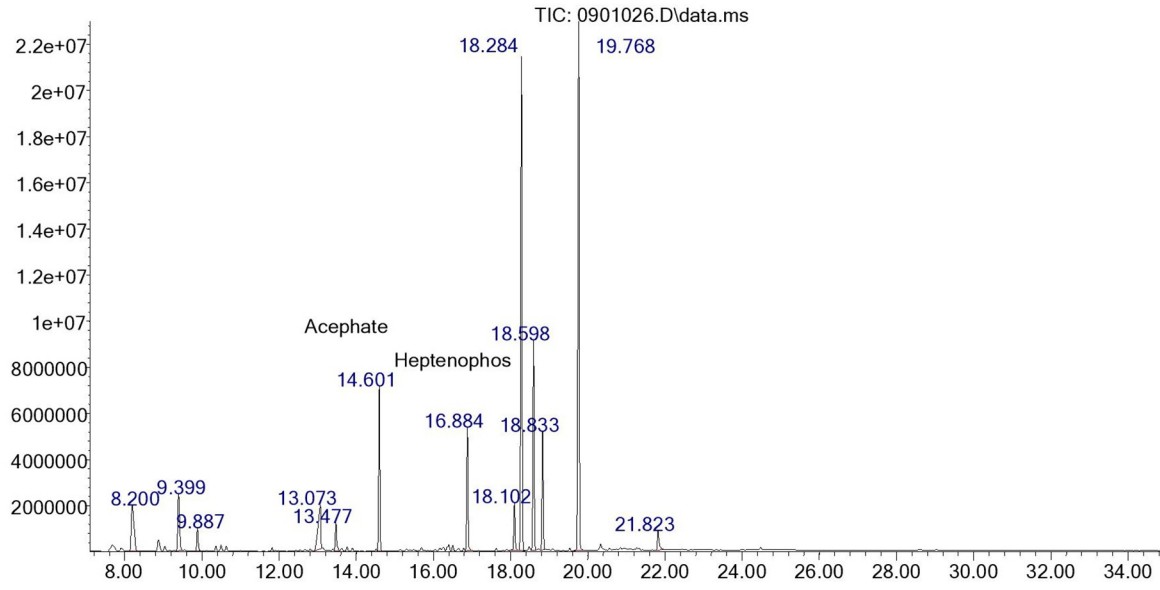

**Fig 3. Chromatogram of heptenophos and acephate in Kale.**

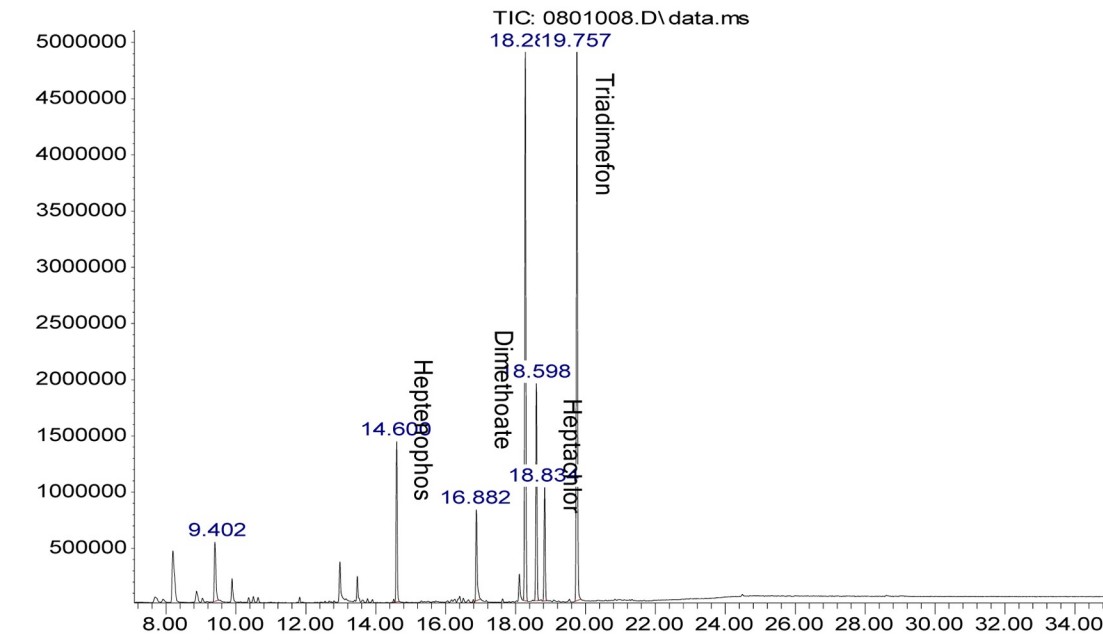

**Fig 4. Chromatogram of triadimefon, dimethoate and heptenophos in tomato.**

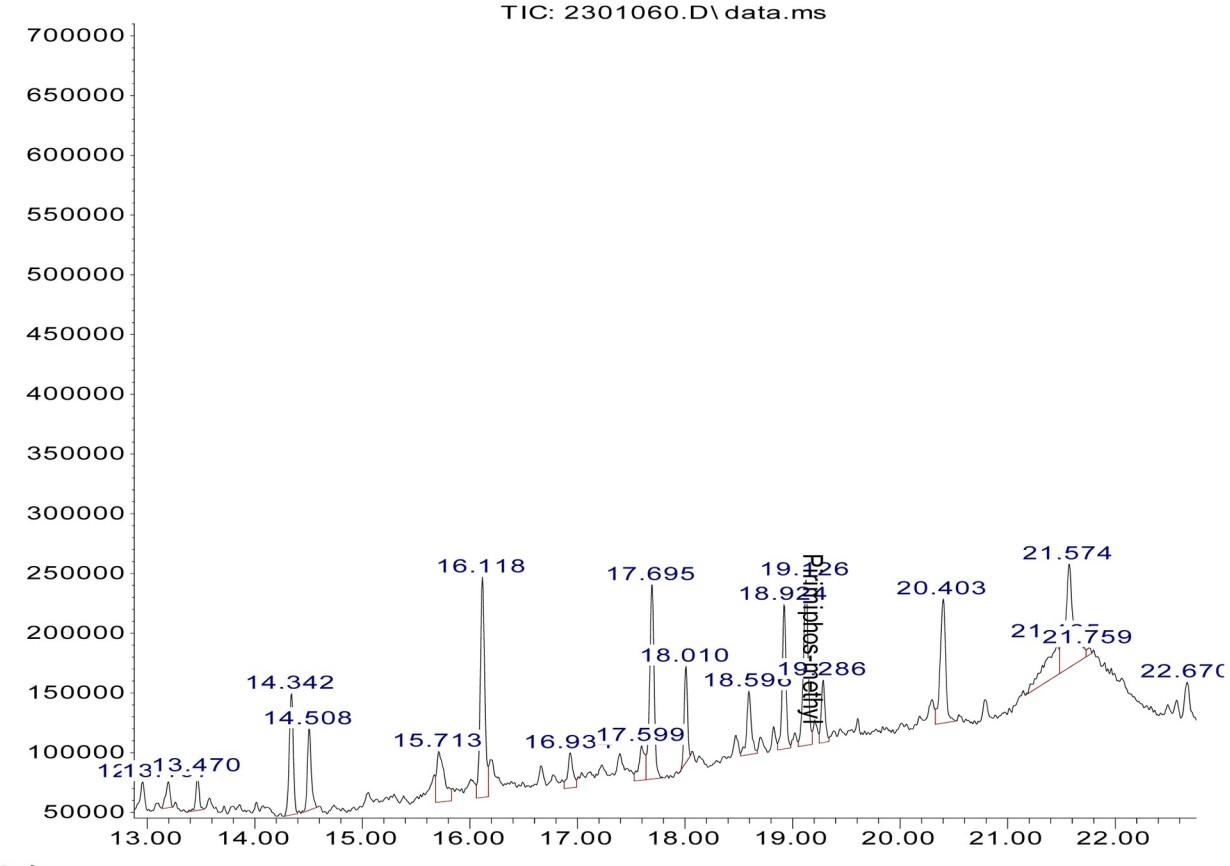

**Fig 5. Chromatogram of pirimiphos-methyl in onion.**

Ethiopian mustard (17.6%), cabbages (16.7%), and tomatoes (8.3%). Owing to increased daily consumption of these fresh vegetables among lower and middle income urban and peri-urban dwellers, the fresh vegetable chain prompts high public health concern.

## Co-occurrence of pesticides residues and bacterial contaminants

Co-contamination of pesticide residues and pathogenic bacteria was noted in the tested samples. A considerable proportion (46.4%) of fresh vegetable samples tested positive for both pesticide residues and bacterial contaminants. Vegetables from farms (60.7%) contained more pesticides and bacterial contaminants than vegetable samples from the market places (41.8%). Chi-square ($\chi^2$) test showed a statistically significant difference (p = 0.010) in the levels of pesticide residues and bacterial contamination between vegetables from farms and markets. The number of bacterial pathogens isolated from a single sample differed significantly among market places (p = 0.022), while the difference was not significant for pesticide residues (p = 0.318) in the same locations. The co-contamination level of fresh vegetables was significantly different in vegetable samples (p = 0.02) with kale (88.2%), onions (64.7%), Ethiopian mustard (63.2%), African nightshade (57.1%) and Chinese cabbage (54.5%), being highly contaminated with both pesticides residues and bacterial contaminants compared with carrots (14.3%), avocadoes (16.7%) and sweet paper (18.8%) as shown in Table 8.

**Table 4. Concentration of pesticides residues detected.**

| Pesticides residues detected | Pesticides residues levels in mg/kg | | |
|---|---|---|---|
| | Minimum | Maximum | Mean |
| Tetramethrin | 0.0329 | 1.3733 | 0.7031 |
| Pirimiphos—methyl | 0.0003 | 1.4093 | 0.6798 |
| Dinoseb acetate | 0.2492 | 0.8961 | 0.5715 |
| Permethrin | 0.0009 | 2.4537 | 0.4760 |
| Endosulfan (Beta) | 0.0008 | 2.3416 | 0.3457 |
| Cabaryl | 0.0215 | 1.5068 | 0.3178 |
| Triadimefon | 0.0011 | 1.6047 | 0.3122 |
| Profenofos | 0.0176 | 2.1377 | 0.2701 |
| Bioallethrin | 0.2542 | 0.2542 | 0.2542 |
| Acephate | 0.0463 | 1.2377 | 0.2515 |
| Toxaphene | 0.0216 | 0.9753 | 0.2503 |
| Cyhalothrine (Lambda) | 0.0686 | 1.4432 | 0.2496 |
| Trichlorform | 0.0001 | 1.4851 | 0.2184 |
| Dichorprop | 0.1625 | 0.1625 | 0.1625 |
| Dimethoate | 0.0004 | 0.9941 | 0.1605 |
| Chlorpyrifos | 0.0004 | 1.2549 | 0.1495 |
| Triadimenol | 0.0120 | 0.4369 | 0.1442 |
| Oxyfluorfen | 0.0031 | 0.4095 | 0.1353 |
| Dieldrin | 0.0011 | 0.5271 | 0.1221 |
| Endosulfan (Alpha) | 0.1087 | 0.1087 | 0.1087 |
| Cyhalothrin (Gamma) | 0.0047 | 0.4416 | 0.1051 |
| Bifenthrin | 0.1030 | 0.1030 | 0.1030 |
| Anilazine | 0.2203 | 0.6203 | 0.1012 |
| Theobromine | 0.0920 | 0.0920 | 0.0920 |
| Malathion | 0.0362 | 0.1921 | 0.0917 |
| Endrin aldehyde | 0.0001 | 0.4237 | 0.0875 |
| Propanil | 0.0018 | 0.0836 | 0.0836 |
| Empethrine | 0.0192 | 0.1538 | 0.0725 |
| Cypermethrin | 0.0572 | 0.0636 | 0.0604 |
| Vamidothion | 0.0086 | 0.2564 | 0.0572 |
| Fenitrothion | 0.0237 | 0.0809 | 0.0523 |
| Methiocarb | 0.0035 | 0.0479 | 0.0479 |
| Heptenophos | 0.0004 | 0.1735 | 0.0397 |
| Captafor | 0.0054 | 0.0661 | 0.0272 |
| Fenthion | 0.0142 | 0.0373 | 0.0257 |
| Bendiocarb | 0.0057 | 0.0624 | 0.0231 |
| Monocrotophos | 0.0226 | 0.0226 | 0.0226 |
| Prarethrin | 0.0214 | 0.0214 | 0.0214 |
| Barban | 0.0001 | 0.0326 | 0.0164 |
| Hexaconazole | 0.0007 | 0.0256 | 0.0132 |
| Metalaxyl | 0.0100 | 0.0100 | 0.0100 |
| Captain | 0.0001 | 0.0231 | 0.0092 |
| Chlorothalonil | 0.0062 | 0.0062 | 0.0062 |
| Carbofuran | 0.0001 | 0.0181 | 0.0031 |
| Fenobucarb | 0.0001 | 0.0001 | 0.0001 |
| Fenothiocarb | 0.0001 | 0.0001 | 0.0001 |

(*Continued*)

**Table 4.** (Continued)

| Pesticides residues detected | Pesticides residues levels in mg/kg | | |
|---|---|---|---|
| | Minimum | Maximum | Mean |
| Binopacryl | 0.0001 | 0.0001 | 0.0001 |
| Flumetralin | 0.0001 | 0.0001 | 0.0001 |
| Quinoclamine | 0.0001 | 0.0001 | 0.0001 |
| Pyriproxyfen | 0.0001 | 0.0001 | 0.0001 |
| Cinerin | 0.0001 | 0.0001 | 0.0001 |
| Oxamyl | 0.0001 | 0.0001 | 0.0001 |

**Table 5. Pesticides residue excess over Codex default limit.**

| | No. of samples analyzed | No. with detectable residues (%) | | Pesticides residues levels in mg/kg | | | Codex default limit | % excess over Codex default limit |
|---|---|---|---|---|---|---|---|---|
| | | | | Minimum | Maximum | Mean | | |
| Onions | 163 | 105 | 64.4 | 0.0001 | 2.4537 | 0.3194 | 0.01 | 96.9 |
| Water melon | 19 | 8 | 42.1 | 0.0001 | 1.3733 | 0.2195 | 0.01 | 95.4 |
| Tomatoes | 180 | 99 | 55.0 | 0.0001 | 1.2549 | 0.2127 | 0.01 | 95.3 |
| Sweet paper | 68 | 37 | 54.4 | 0.0001 | 1.5068 | 0.1529 | 0.01 | 93.5 |
| Chinese cabbage | 26 | 10 | 38.5 | 0.0016 | 0.3198 | 0.0712 | 0.01 | 86 |
| Cucumber | 13 | 3 | 23.1 | 0.0008 | 0.1921 | 0.0678 | 0.01 | 85.3 |
| African nightshade | 19 | 3 | 15.8 | 0.0004 | 0.103 | 0.0592 | 0.01 | 83.1 |
| Carrots | 16 | 4 | 25.0 | 0.0019 | 0.2492 | 0.0457 | 0.01 | 78.1 |
| Amaranths | 7 | 2 | 28.6 | 0.0424 | 0.0424 | 0.0424 | 0.01 | 76.4 |
| Kale | 22 | 5 | 22.7 | 0.0006 | 0.1476 | 0.0407 | 0.01 | 75.4 |
| Ethiopian mustard | 7 | 1 | 14.3 | 0.0335 | 0.0335 | 0.0335 | 0.01 | 70.1 |
| Egg plant | 13 | 5 | 38.5 | 0.0026 | 0.0565 | 0.0284 | 0.01 | 64.8 |
| Green beans | 7 | 1 | 14.3 | 0.0146 | 0.0192 | 0.0169 | 0.01 | 40.9 |
| Cabbage | 19 | 5 | 26.3 | 0.0005 | 0.0215 | 0.0123 | 0.01 | 18.4 |
| Okra | 14 | 3 | 21.4 | 0.0018 | 0.0171 | 0.0095 | 0.01 | |

Binary logistic regression analysis showed that pesticide contamination of fresh vegetables was more likely to induce bacterial contamination. Vegetables with pesticide residues were 2.231 times more likely to be contaminated with bacteria contaminants (OR: 2.231; 95% CI: 0.501, 8.802). Likewise, the use of the same wiping cloth/towel in cleaning fresh fruits increased the likelihood of contaminating fresh vegetables with both pesticides and bacterial contaminants by 29% (OR: 1.288; 95% CI: 0.251). The location of vegetables with respect to farm and

**Table 6. Levels of pesticides residues by sampling sites.**

| Variable | | Sampling place | | | | | | Total | |
|---|---|---|---|---|---|---|---|---|---|
| | | Farm | | Market | | Highway | | n | % |
| | | n | % | n | % | n | % | | |
| Level of pesticides residues with reference to Codex MRL standards | Above Codex MRL standards | 130 | 71.0 | 189 | 74.7 | 27 | 90.0 | 346 | 74.2 |
| | Below Codex MRL standards | 53 | 29.0 | 64 | 25.3 | 3 | 10.0 | 120 | 25.8 |
| Total | | 183 | 100.0 | 253 | 100.0 | 30 | 100.0 | 466 | 100.0 |

**Table 7. Types of pathogenic bacteria isolated from fresh vegetables.**

| Variable | | Sampling location | | | | Total | |
|---|---|---|---|---|---|---|---|
| | | Farm | | Market | | N | % |
| | | n | % | n | % | | |
| Types of microbes | Enterobacter | 9 | 37.5 | 70 | 59.3 | 79 | 55.6 |
| | Pseudomonas aeruginosa | 8 | 33.3 | 38 | 32.2 | 46 | 32.4 |
| | E.coli | 11 | 45.8 | 29 | 24.6 | 40 | 28.2 |
| | Citrobacter | 3 | 12.5 | 35 | 29.7 | 38 | 26.8 |
| | Klebsiella oxytoca | 1 | 4.2 | 20 | 16.9 | 21 | 14.8 |
| | Salmonella | | | 11 | 9.3 | 11 | 7.7 |

**Table 8. Co-occurrence of pesticides residues and bacterial contaminants.**

| Variable | | Sampling location | | | | Total | |
|---|---|---|---|---|---|---|---|
| | | Farm | | Market | | n | % |
| | | n | % | n | % | | |
| Contain both pesticides residues and bacterial contaminants | No | 24 | 39.3 | 110 | 58.2 | 134 | 53.6 |
| | Yes | 37 | 60.7 | 79 | 41.8 | 116 | 46.4 |
| Total | | 61 | 100.0 | 189 | 100.0 | 250 | 100.0 |

**Table 9. Binary logistic regression of factors associated with pesticides and bacterial contamination of fresh vegetables locally produced and consumed.**

| | B | Wald | df | Sig. | Exp(B) | 95.0% C.I for EXP(B) | |
|---|---|---|---|---|---|---|---|
| | | | | | | Lower | Upper |
| Pesticides contamination | 3.484 | 4.379 | 1 | 0.036 | 2.231 | 0.501 | 8.802 |
| Storage | 0.776 | 0.917 | 1 | 0.338 | 2.172 | 0.444 | 10.622 |
| Water used in irrigation | 0.209 | 0.028 | 1 | 0.868 | 1.234 | 0.104 | 14.656 |
| Safe use train | 1.117 | 0.513 | 1 | 0.474 | 3.054 | 0.144 | 64.796 |
| Location | -4.030 | 5.307 | 1 | 0.021 | 0.018 | 0.112 | 0.548 |
| Use same wiping cloth | 0.253 | 0.092 | 1 | 0.016 | 1.288 | 0.251 | 6.610 |
| Splash water | -0.440 | 0.232 | 1 | 0.630 | 0.644 | 0.107 | 3.868 |

market place influenced the likelihood of co-contamination of fresh vegetables. The likelihood of co-contamination was 1.8% less for vegetables from the markets compared with those from farms (OR: 0.018; 95% CI: 0.112, 0.548). Other factors, including water used for irrigation, storage, attending pesticides, safe use and hygienic handling of vegetables, and splashing water to freshen vegetables did not significantly influence the co-contamination of fresh vegetables (Table 9).

## Discussion

### Pesticides residue contamination

Locally produced and consumed vegetables are highly contaminated with pesticide residues. Continuous consumption of contaminated vegetables can lead to the accumulation of toxic substances in the body causing long term health effects [10,25,26]. Organophosphorus followed by organochlorines, pyrethroids and carbamates were the most pesticides detected in

most of the samples analyzed. Similar results were reported in Ghana, where organochlorine followed by organophosphorus and synthetic pyrethroid pesticides where detected in analyzed samples of fresh fruits and vegetables [27]. Organophosphate pesticides, which constituted a large proportion of pesticide residues detected, are endocrine disruptors and account for most health effects of exposure [28,29]. Considering the levels of pesticide residues detected in vegetable samples, consumers are at risk of dietary exposure. Exposure to organophosphate pesticides had been associated with decrease in the activity of acetyl cholinesterase and alterations in the level of haematological parameters, liver and renal dysfunctions [30].

From the current study, the prevalence of pesticide contamination (47.5%) is lower than the previously reported level (95.8%) from Dar es Salaam markets [31]. This may be due to relatively larger coverage and sample size used as opposed to the Dar es Salaam study, where only tomatoes and watermelons samples were analyzed. Results have revealed that 73.7% of all vegetable samples from Dar es Salaam had detectable levels of pesticide residues. High levels of pesticides in Dar es Salaam can be explained by multiple sourcing of vegetables. Sourcing of fresh fruits and vegetables long the highways which recorded highest levels of pesticide residues may also contribute to increased proportion of samples with pesticides residues.

Presence of pesticides residues in vegetables had also been reported in other countries [32,33], but the proportion of samples with detectable levels of pesticides in this study are higher than those reported in other developing countries [10,32,34], and much higher than levels of pesticides residues detected from the E.U. and U.S. pesticides monitoring programs [12,13], where the average pesticides residues exceeding established MRL standards were below 2%. This may be attributed to poor knowledge of pesticide's safe use and weak institutional frameworks on the monitoring and judicious pesticide use.

A total of 52 different types of pesticide residues were detected from all fresh fruits and vegetable samples. This number is much higher than that reported from other countries [10,32,34]. For instance, only 22 pesticide residues were detected from vegetable samples from China [32]. Multiples residues were evident in this study. Up to seven pesticides residues were detected in a single sample. Similar findings (multiple residues) had also been reported from both developed and developing countries [9,10,12,32], but the number and proportions were much lower compared to this study, except proportions from Brazil where samples reported up to 10 different residues [35].

Samples collected from high ways had a relatively high proportion of samples with more pesticides than samples collected from farms and market places, suggesting higher concentrations of pesticides sprayed just before harvesting. This might also be attributed to observed secondary treatment of vegetables at collection centres and farmers over spraying ready-to-harvest vegetables in efforts to increase shelf-life and customer attraction. Failure to observe the pre-harvesting intervals and injudicious use of pesticides may account for this as well [2].

As evidenced from this and other studies that the most common chemical family of pesticides used in horticultural production were organophosphorus [36–38], 95.2% of all pesticides residues detected were organophosphorus as well. This is contrary to the findings from Brazil where only 30.8% of all pesticides residues were organophosphorus [35] and India where none of organophosphate pesticides except traces of malathion and chlorpyrifos residues in mango fruits [11]. These findings indicate that almost all that goes in during production (pesticide spraying) comes out as pesticide residues after harvests. Some samples contained unauthorised pesticide active substances. Organochlorine pesticides were detected in quantifiable levels, but these pesticides had been banned for agricultural use in Tanzania. Similar findings were reported in the U.S. and Ghana [9,27], where the use organochlorine pesticides had been banned many years ago. Their presence in food samples suggests some illegal business and/or

their persistence in the food chain systems because most leafy vegetables tend to absorb organochlorine pesticide residues from the soils and translocate them into edible crop tissues [9].

The concentration of pesticides residues were compared with the FAO/WHO Codex Alimentarius Commission maximum residue limits (MRLs) [8]. High proportion (74.2%) of fresh fruits and vegetables had concentrations of pesticide residues above the MRLs. A similar study in Ghana revealed that 32.8% of the fruit samples analyzed contained residues above the accepted MRL [26]. The proportion from this study is far above those reported in the E.U. countries, where 97.4% of the tested food samples fell within the legal limits [12], Egypt where 81.5% did not have detectable levels of pesticides residues [33], Brazil [35] and Pakistan where only 3% exceeded MRLs [39] and Turkey where 9.2% of all fresh fruits and vegetable contained pesticide residues above MRLs [40].

Vegetables produced under smallholder horticultural production therefore poses high public health risks and increased risks of detrimental health effects among consumers as well as farmers who are primary consumers of these vegetables. Excessive pesticide residues and their metabolites in dietary constituents and food materials have been linked to chronic health effects, imparting child development. Alterations in birth weight, head circumference, retarded brain development, mid-arm circumference and ponderal index of the neonates are all linked to pesticides exposure [41]. The general population, including children are therefore at high risk of dietary exposure and resultant health effects resulting from consumption of contaminated fresh fruits and vegetables.

Generally, tomatoes, onion, sweet paper, water melons, cabbage and Chinese cabbage recorded high concentration levels of pesticide residues. Similar findings were reported from Ghana [36] though only in small proportion of tomatoes and pepper had levels above MLRs. These concentration levels are much higher than those reported in China [34] and elsewhere [12,13,32]. This signifies significant food safety challenge in smallholder vegetable production, indiscriminate pesticides use as well as the poor quality of pesticides used and supplied in the local market. Some indigenous species of local vegetables did not have FAO/WHO reference MRL values from the FAO Codex MRL standards. Consequently, they recorded high excess levels of pesticide residues over a default codex maximum pesticide residue limit of 0.01mg/kg of food sample, thereby increasing exposure risks and monitoring challenges of pesticides residues in locally produced vegetables.

Among all pesticide residues detected, tetramethrin, pirimiphos-methyl, permethrin, endosulfan (beta), and carbaryl recorded high mean concentration in fresh fruits and vegetable samples. Other pesticide residues with higher means concentrations included profenofos, chloropyrifos, bioallethrin, acephate, toxaphene, cyhalothrine (lambda) and trichlorform. These pesticides had been reported to be used in smallholder vegetable production systems in Tanzania [2,4,37,42,43], indicating that continuous accumulation of pesticides residues in the environment could also contribute to the levels of pesticides residues. Similar pesticides were reported in Ghana [27], but the concentration levels detected in this study are much higher compared with previous studies [32,44], signifying that dietary exposure may emanate from consumption of fresh fruits and vegetable produced by smallholder farmers. Analysis of pesticide residues in fruits and vegetables from the Aegean region in Turkey showed a progressive decrease in samples that exceeded the MRLs over a period of three years due to tighter government regulations on pesticides use, increased education of farmers, and effective control of pesticide business and implementation of integrated pest management methods [40]. Therefore, higher concentration of pesticides residues may therefore be explained by poor pesticides use, use of highly hazardous pesticides, weak enforcement of pesticides regulations by the government, limited access to pesticides safe use extension education and overreliance on synthetic chemical pesticides in smallholder vegetable production systems [2,37,45].

## Bacterial contamination

Fresh fruits and vegetables produced by smallholder farmers have been contaminated with fae-cal and other contaminants. A considerable high proportion (63.2%) of samples tested was contaminated with at least one bacterial pathogen. This prevalence in higher than the rate reported in Ethiopia where only 48.7% were positive for bacterial contamination [7]. Patho-genic bacteria including *E. coli* and *Salmonella spp* were isolated from fresh vegetables, con-trary to the study in Brazil which did not report *Salmonella spp.* in vegetable samples [46]. *Salmonella* are important pathogens for public health. Their association with food borne dis-ease outbreaks and economic burden has been reported [47]. Presence of these microorgan-isms in vegetable samples provides critical economic and public health concerns [48].

Moreover, *Pseudomonas aeruginosa*, *Citrobacter*, *Enterobacter and Klebsiella oxytoca* were also among the identified microbial species in tested vegetables. Among these six identified bacterial species; *Enterobacter* (55.6%) was the commonest contaminant contrary to the previ-ous study in Ethiopia where *E. coli* (31.4%) was the commonest contaminant. These bacterial contaminants are commonly isolated from faecal and urine samples responsible for a wide range of gastrointestinal disorders [24]. These findings, therefore, signify that smallholder fresh fruits and vegetables pose high public health concerns among consumers.

Samples from farms recorded higher *E. coli* contamination compared to those from mar-kets, while *Salmonella spp.* were isolated from samples collected from market places, only indi-cating that vegetable vendors may play a key role in second-hand contamination of fresh vegetables. The production phase, therefore, constitutes the main contamination point of fae-cal contaminants of fresh vegetables because *E. coli* is a faecal coliform bacterium that is nor-mally excreted in stool [7]. High risks of faecal contamination may have emanated from people reported to be entering and/or urinating/defecating in the farms. Fertilizers, irrigation water, wild animal intrusion, insects, pesticides/fungicides, crop debris, and flooding area also potential sources of microbial contamination at production level [47]. Contaminated water splashed on vegetables during selling, and possible contamination of tap water with sewage drainage systems at the market places may account for the market contaminations.

Kale, cabbage, spinach, Ethiopian mustard, water splashed to freshen vegetables, nightshade amaranths and watermelons had the highest bacterial contamination. Likewise, tomatoes, onions carrots, ripe banana and Chinese cabbage had considerable high pathogenic contami-nations while the least contaminated were okra and avocadoes. These findings are similar to the previous studies, but only cabbage was highly contaminated [7]. High contamination of these leafy vegetables emanates from the large surface area, contaminated water splashed to freshen, and possible unhygienic handling by farmers and vegetable vendors as well as custom-ers touching during selection of vegetables. The unhygienic market condition may also account for increased vegetable contaminations. Moreover, high levels of *E. coli* had been found in irrigation water [49] which may also account for increased microbial contamination of vegetables from the farms.

Higher prevalence of *Enterobacter* and detection of *salmonella* from market places may be due to skin contact from customers touching vegetables and environmental contamination [50]. Likewise, cross-contamination that occurs during transportation and multiplication microorganisms after harvesting of leafy vegetables and storage temperature conditions may also account for this as well [51]. Prevalence of some bacterial contaminants from this study was lower than those reported from other countries. The prevalence of salmonella (7.7%) is much lower than that reported (24%) from Ethiopia [50]. Likewise, the prevalence of *Escheri-chia coli* (28.2%) was lower than those reported in 53.1% samples from Brazil [52]. These path-ogens (*E. coli* and *Salmonella spp.*) had been isolated from the vegetable vendors [3], hence

these findings support hypothesis that fresh vegetables vendors are potential sources of pathogenic *Salmonella* and *E. coli*. Likewise, market-related handling, especially where provision for better sanitary standards are inadequate are also reported to be the main source of contamination [1].

Kale, carrots, sweet paper, and watermelons were highly contaminated with *enterobacter*, while avocado samples, cabbage, okra, amaranths, spinach, and tomatoes were contaminated with *Pseudomonas aeruginosa*. Furthermore, Ethiopian mustard, nightshade, and Chinese cabbage were contaminated with *E.coli*. *Salmonella spp*. were isolated from onion, amaranths, Chinese cabbage, and water samples. Owing to increased daily consumption of these fresh vegetables among lower and middle-income dwellers in urban and peri-urban settings where locally produced vegetables are consumed, the fresh vegetables supply chain prompts high public health concern.

The isolated genera of *Escherichia*, *Klebsiella*, *Enterobacter*, *Salmonella*, and *Citrobacter*, constitute opportunistic pathogens responsible for a wide range of infections. Similar pathogenic microbes had been isolated in faecal and diarrhoea samples [24], indicating faecal contamination of fresh vegetables. Human factors, including unhygienic practices and deliberate contamination also account for bacterial contamination of fresh vegetables [3]. *E coli* is the major cause of diarrhoea and urinary tract infections, including prostatitis and pyelonephritis. These pathogens cause a range of illnesses, including urinary tract, infections, respiratory tract, skin, soft-tissues, joints, bones, eyes and the Central Nervous System (CNS) [49]. Consumers of fresh vegetables are therefore at risk of infection from these diseases and other food borne diseases (FBD), including salmonellosis and cholera.

## Co-contamination of fresh vegetables

Vegetables produced by smallholder farmers had been co-contaminated with pesticide residues and pathogenic bacteria. Pearson correlation test showed a positive, non-statistically significant correlation (r = 0.103, p = 0.2) between the levels of pesticide and bacterial contamination. From the present study, a considerable proportion (46.4%) of fresh vegetables contained both pesticide residues and bacterial contaminants. Vegetables from farms (60.7%) were highly contaminated with both. Statistically significant difference (p = 0.010) in co-contamination levels of vegetables from farms and markets indicates high health risks to farmers who are both producers and primary consumers of vegetables. There was a significant difference in co-contamination levels among the vegetable samples (p = 0.02) with kale, onions, Ethiopian mustard, nightshade, and Chinese cabbage being highly contaminated.

Binary logistic regression analysis showed the association between pesticide residues and bacterial contamination of vegetables. Pesticides residues were more likely to induce bacterial contamination. Excessive pesticides reported among smallholder farmers in Tanzania [2] may account for the increased microbial contamination. This is in support of the hypothesis that pesticide chemical composition can act as a stimulatory or inhibitory substrate for microbial growth [18]. Likewise, pesticide solutions sprayed on agricultural crops in controlling pests and insects, mostly with organophosphorus and carbamates as the active ingredients had been reported to provide a suitable environment for the survival and growth of human pathogenic microbes, including *E. coli*, and *Salmonella* [49].

Unhygienic handling of vegetables, such as the use of the same wiping cloth/towel in cleaning fresh fruits, increased the likelihood of contaminating fresh vegetables with both pesticides and bacterial contaminants.

The co-occurrence of pesticide residues and bacterial contaminants from this study indicates the symbiotic existence of pesticides and microbial activities in the environment. Several

bacterial and fungal isolates had been reported to have a significant ability to carry out the degradation of pesticides. Among many bacterial strained identified to be resistant pesticides are bacterial belonging to genus *Enterobacter* and *Pseudomonas* [53]. Likewise, *Bacillus thuringiensis* had been found to able to degrade malathion while *Stenotrophomonas maltophilia* and *Rhodococcus erythropolis* are to be responsible for the majority of the degradation of pesticides [17,53]. The presence of *Enterobacter* and *Pseudomonas* bacteria from pesticides contaminated fresh vegetables further suggests the effects of pesticides use on the growth of pathogenic bacteria in the environment.

The results of this study show that the contamination levels of pesticide residues and bacterial contaminants could be perceived as a serious problem as most fresh fruits and vegetables recorded values of pesticide residues far above the MRLs with pathogenic bacteria isolated in higher proportions. Pesticides that exceeded the MRLs was higher in most vegetables that are consumed after raw or semi-cooked such as watermelons, carrots, cucumber, tomatoes, onion and sweet paper.

Certain operations, such as washing, cooking and peeling are reported to reduce pesticide residues [54]. Effective, proper and multiple washing of vegetables while changing the washing water and/or peeling had been significantly found to reduce levels of pesticides in ready to eat vegetables while poor practices in washing result in the occurrence of pesticide residues in vegetables [4]. These processes, which are the most common forms of processing and preliminary steps in vegetable preparation had been reported to be effective processes for reduction of pesticide residues in fresh fruits and vegetables [55]. Furthermore, washing of vegetables with water and with chlorine had been reported to reduce bacterial counts on vegetables [56]. On contrary, a study in India reported that contaminated water, poor domestic hygiene, and peeling of fruits were the sources of contamination of fresh juice produced by vendors from fresh fruits [57]. Therefore, though these processes are found to effectively reduce levels of pesticides residues and bacterial contamination, determining the effectiveness of the reduction of pesticides and consequently bacterial contaminants by these procedures demands an understanding of whether systemic or contact pesticides residues dominate the chemical contamination of fresh fruits and vegetables as well and the hygienic conditions under which processing is done. Owing to poor household handling and processing of vegetables and the levels of pesticides residues exceeding MRLs, washing and peeling may not result in significant removal of pesticides residues and bacterial contaminants in fresh fruits and vegetables.

## Conclusion

In conclusion, this study is probably for the first to assess factors contributing to the co-occurrence of pesticide residues and bacterial contamination in fresh vegetables under smallholder horticultural production systems in Tanzania. The results of this study provide significant information concerning pesticide residues and bacterial contamination in fruits and vegetables. According to the results, organophosphorus, organochlorines, carbamates, and pyrethroids residues dominated vegetable samples. The most detected pesticides active ingredients were oxyfluorfen, cyhalothrine (lambda and gamma), profenofos, triadimenol, chlorpyrifos, triadimefon, endosulfan (beta), carbofuran and dieldrin while *Enterobacter E. coli*, *Salmonella*, *Pseudomonas aeruginosa*, *Citrobacter*, *Klebsiella oxytoca*, and *Salmonella* were isolated.

A significant association was found between pesticide residues and bacterial contamination. The presence of pesticide residues in fresh vegetables influenced bacterial contamination, signifying the effects of pesticide use on bacterial contamination of fresh vegetables. General lack of adherence to both good agricultural practices (GAPs) and good hygienic practices (GHP) at

the production and marketing nodes of smallholder vegetable supply chains may account for this.

In this study, the percentage of pesticide residues exceeding the MRLs were far above those reported elsewhere. Based on this rate, pesticides residues signify possible human risks. Moreover, onions, watermelons, tomatoes, and sweet paper had more pesticide residues above the CODEX MRLs while kale, cabbage, spinach, Ethiopian mustard, nightshade, amaranths, watermelons tomatoes, onions ripe banana, and Chinese cabbage were highly contaminated with bacteria contaminants.

It is suggested that monitoring programs should be encouraged so as to generate data for managing pesticides residues in fresh fruits and vegetables. More strict measures such as developing pesticides monitoring and surveillance systems at farmer level, sustainable vegetable and food safety control program, educating farmers and promoting the use of greener pesticides to mitigate the health effects of pesticides and bacterial contaminants is of paramount importance. Further studies on the antibiotic resistance of identified bacterial strains and analysis of levels of pesticide residues in the blood and urine of the exposed person are therefore warranted.

## Supporting information

**S1 Data.**
(DOC)

**S1 Dataset.**
(DOC)

**S2 Dataset.**
(XLS)

## Acknowledgments

The research team acknowledges Tropical Pesticides Research Institute (TPRI), for providing pesticide standards of technical grade from certified registrants, GC-MS, technical assistance and laboratory facilities for pesticide residues and bacterial contamination tests.

## Author Contributions

**Conceptualization:** Jones A. Kapeleka.

**Data curation:** Jones A. Kapeleka.

**Formal analysis:** Jones A. Kapeleka.

**Funding acquisition:** Jones A. Kapeleka.

**Methodology:** Jones A. Kapeleka.

**Supervision:** Elingarami Sauli, Omowunmi Sadik, Patrick A. Ndakidemi.

**Writing – original draft:** Jones A. Kapeleka.

**Writing – review & editing:** Elingarami Sauli, Omowunmi Sadik, Patrick A. Ndakidemi.

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
