## [Decision Letter · Decision Letter 0]

16 Apr 2020

PONE-D-20-01490

Co-exposure risks of pesticides residues and bacterial contamination in fresh fruits and vegetables under smallholder horticultural production systems in Tanzania

PLOS ONE

Dear Jones Kapeleka,

Thank you for submitting your manuscript to PLOS ONE. After careful consideration, we feel that it has merit but does not fully meet PLOS ONE’s publication criteria as it currently stands. Therefore, we invite you to submit a revised version of the manuscript that addresses the points raised during the review process.

We would appreciate receiving your revised manuscript by 25-05-2020. To enhance the reproducibility of your results, we recommend that if applicable you deposit your laboratory protocols in protocols.io, where a protocol can be assigned its own identifier (DOI) such that it can be cited independently in the future. For instructions see: http://journals.plos.org/plosone/s/submission-guidelines#loc-laboratory-protocols

We look forward to receiving your revised manuscript.

Kind regards,

Ch Ratnasekhar, Ph.D.

Academic Editor

PLOS ONE

Journal Requirements:

Please ensure that your manuscript meets PLOS ONE's style requirements, including those for file naming. The PLOS ONE style templates can be found at http://www.plosone.org/attachments/PLOSOne_formatting_sample_main_body.pdf and http://www.plosone.org/attachments/PLOSOne_formatting_sample_title_authors_affiliations.pdf

Reviewers' comments:

Reviewer's Responses to Questions

**Comments to the Author**

1. Is the manuscript technically sound, and do the data support the conclusions?

Reviewer #1: Partly

Reviewer #2: Partly

2. Has the statistical analysis been performed appropriately and rigorously? 

Reviewer #1: Yes

Reviewer #2: No

3. Have the authors made all data underlying the findings in their manuscript fully available?

Reviewer #1: Yes

Reviewer #2: Yes

4. Is the manuscript presented in an intelligible fashion and written in standard English?

Reviewer #1: No

Reviewer #2: No

5. Review Comments to the Author

Reviewer #1: Comments:

The manuscript entitled “Co-exposure risks of pesticides residues and bacterial contamination in fresh fruits and vegetables under smallholder horticultural production systems in Tanzania” is well written and innovative in its aims to investigate the effect of simultaneous exposure of pesticide residues and bacterial contaminants in food matrices. The aim constraint with the manuscript being accepted for publication lies on the fact that the authors have investigated large number of samples set for multiple group of pesticides as well as bacterial contaminants which seems to be statistically relevant. However, there are few queries and comments.

Please increase the resolution of each figures and some of tables can be merged giving better explanation of the co-exposure of the pesticides and bacterial contaminants.

The manuscript needs major revision in terms of the comments provided below.

1. Abstract

• From Page 1-3, The whole content in the abstract has been written in duplex. Please correct it.

• Page 1, Line “Vegetables from farms (60.7%) contained more dual contaminants than market-based vegetables” …Any logical explanation behind this?

• Page 2, Line “Immediate actions are therefore required to mitigate the health effects of these toxic and bacterial contaminants” …Specify the action to be taken.

2. Introduction

• Page 3, Line “However, one of the health threats is unacceptable pesticide residue levels and bacterial contamination” …. Reframe this sentence

• Page 4, Line “Exposure to pesticides is, therefore, more likely to affect the general population through consumption of contaminated food with pesticide residues (Neff et al., 2012) …. Please use same reference citation pattern as done in whole manuscript.

• Page 4, Line, “as pesticides use had been linked to support and increase microbial growth in vegetable produce” Rewrite the sentence.

• Page 4, Line “However, there is limited information on the co-exposure risks from pesticide residues and pathologic bacterial contaminants in smallholder horticultural production systems in Tanzania” …. Please highlight briefly in the introduction section about the earlier report of co-exposure of pesticides and bacterial contaminants.

3. Materials and Methods

• Page 5, section: samples…...What criteria were followed for sample selection and why only 250 samples out of 613 were selected for bacterial contaminants study.

• Please add a couple of section in material and method section such as chemicals and reagent stating the source of the chemicals/reagents purchased. Also add section regarding the standard preparation for pesticides residues.

• Change the title of section “Sample analysis”

• “Recovery, quantitative evaluation and detection limits” Move this section to Results and discussion and elaborate in detail.

• Mention a section for the tools used for the statistical and data analysis

 

4. Results

• Change the title “Horticultural samples collected”

• Change the title “Nature of pesticides used in vegetable production” to class of pesticides…

• This whole section needs language correction

5. Discussion

• This section can be shorten excluding the repeated lines already discussed in the introduction section.

Please add number to each section and sub sections throughout the manuscript.

Reviewer #2: Comments on the manuscript entitle “Co-exposure risks of pesticides residues and bacterial contamination in fresh fruits and vegetables under smallholder horticultural production systems in Tanzani”

Comments

1. The abstract needs restructured as the whole abstract repeated again from line no. 20 “This study was carried out to assess co-exposure…………………”.

2. A detailed method of sample preparation is needed. Usually people consume the vegetable after through washing then why you selected unwashed sample?

3. The procedure of quantitative evaluation of pesticide residue is not clear and it needs explanation with the concentration used.

4. In the result section, the presentation of data is in the very poor form, it need extensive revision and restructure. Most of the data represented in percentage only, the statistical tool needs to apply for the significance of study.

5. Studies have also been reported that microbial fauna in soil become adversely affected through the use of pesticide. How would you correlate it with your present study?

6. The discussion section needs revision to incorporate the other relevant studies and their association with your results.

7. Do you have measured the levels of pesticide residues in the blood and urine of the exposed person? If a person consume the vegetable after washing it properly or by other treatment then what will the effect on pesticide residues and bacterial contamination?

8. The recent references including the pesticide induced adverse health effects in agriculture workers such as “Kori et al 2019 and 2020, Journal of Biochemical and Molecular Toxicology” and others can be included in the manuscript for better making the discussion better.

6. PLOS authors have the option to publish the peer review history of their article (what does this mean?). If published, this will include your full peer review and any attached files.

Reviewer #1: Yes: Dr Rakesh Roshan Jha

Reviewer #2: Yes: Rajesh Singh Yadav

---

## [Author Response · Author response to Decision Letter 0]

29 May 2020

All responses are uploaded as separate file and labeled 'Response to Reviewers'

---

## [Decision Letter · Decision Letter 1]

15 Jun 2020

Co-exposure risks of pesticides residues and bacterial contamination in fresh fruits and vegetables under smallholder horticultural production systems in Tanzania

PONE-D-20-01490R1

Dear Dr. Jones Kapeleka,

We’re pleased to inform you that your manuscript has been judged scientifically suitable for publication and will be formally accepted for publication once it meets all outstanding technical requirements.

Kind regards,

Ch Ratnasekhar, Ph.D.

Academic Editor

PLOS ONE

Additional Editor Comments (optional):

Reviewers' comments:

Reviewer's Responses to Questions

**Comments to the Author**

1. If the authors have adequately addressed your comments raised in a previous round of review and you feel that this manuscript is now acceptable for publication, you may indicate that here to bypass the “Comments to the Author” section, enter your conflict of interest statement in the “Confidential to Editor” section, and submit your "Accept" recommendation.

Reviewer #1: All comments have been addressed

Reviewer #2: All comments have been addressed

2. Is the manuscript technically sound, and do the data support the conclusions?

Reviewer #1: Yes

Reviewer #2: Yes

3. Has the statistical analysis been performed appropriately and rigorously? 

Reviewer #1: Yes

Reviewer #2: Yes

4. Have the authors made all data underlying the findings in their manuscript fully available?

Reviewer #1: Yes

Reviewer #2: Yes

5. Is the manuscript presented in an intelligible fashion and written in standard English?

Reviewer #1: Yes

Reviewer #2: Yes

6. Review Comments to the Author

Reviewer #1: (No Response)

Reviewer #2: Comments on the Manuscript entitled "Co-exposure risks of pesticides residues and bacterial contamination in fresh fruits and vegetables under smallholder horticultural production systems in Tanzania".

The authors have addressed the issues raised by the Reviewer.

7. PLOS authors have the option to publish the peer review history of their article (what does this mean?). If published, this will include your full peer review and any attached files.

Reviewer #1: No

Reviewer #2: Yes: Dr. Rajesh Singh Yadav

---

## [Editor Report · Acceptance letter]

17 Jun 2020

PONE-D-20-01490R1 

Co-exposure risks of pesticides residues and bacterial contamination in fresh fruits and vegetables under smallholder horticultural production systems in Tanzania 

Dear Dr. Kapeleka:

I'm pleased to inform you that your manuscript has been deemed suitable for publication in PLOS ONE. Congratulations! Your manuscript is now with our production department. 

Kind regards, 

on behalf of

Dr. Ch Ratnasekhar 

Academic Editor

PLOS ONE